# Classification of Medical Text in Small and Imbalanced Datasets in a Non-English Language

**Vincent Beliveau**[1,2]                                   VINCENT.BELIVEAU@NRU.DK
**Helene Kaas**[1,3]                                          HELENE.KAAS@NRU.DK
**Martin Prener**[1,3]                                        MARTIN.PRENER@NRU.DK
**Claes N. Ladefoged**[4,5]                       CLAES.NOEHR.LADEFOGED@REGIONH.DK
**Desmond Elliott**[6]                                             DE@DI.KU.DK
**Gitte M Knudsen**[1,7]                                            GMK@NRU.DK
**Lars H. Pinborg**[1,3,7]                                  LARS.PINBORG@NRU.DK
**Melanie Ganz**[1,6]                                         MELANIE.GANZ@NRU.DK

[1] *Neurobiology Research Unit, Rigshospitalet, Copenhagen, DK*

[2] *Department of Neurology, Medical University of Innsbruck, Innsbruck, Austria*

[3] *Epilepsy Clinic, Department of Neurology, Rigshospitalet, Copenhagen, DK*

[4] *Department of Clinical Physiology, Nuclear Medicine and PET, Rigshospitalet, Copenhagen, DK*

[5] *Department of Applied Mathematics and Computer Science, Technical University of Denmark, Kongens Lyngby, DK*

[6] *Department of Computer Science, University of Copenhagen, Copenhagen, DK*

[7] *Department of Clinical Medicine, University of Copenhagen, Copenhagen, DK*

**Editors:** Accepted for publication at MIDL 2024

## Abstract

Natural language processing (NLP) in the medical domain can underperform in real-case applications involving small datasets in a non-English language with few labeled samples and imbalanced classes. We evaluated a range of state-of-the-art NLP models on datasets representing this situation and found that current approaches are not sufficiently accurate to allow for fully automated classification, but can potentially be used to filter and reduce the amount of manual labeling.

**Keywords:** NLP, radiology reports, classification

## 1. Introduction

The increasing access to electronic health records (EHR) has opened unparalleled opportunities for big data in the medical domain. However, the information contained in EHR is largely unstructured or semi-structured, and further processing is required to obtain the desired information. In this context, a prominent recurring task is the extraction of relevant labels from medical texts associated with external data. This has been particularly relevant in radiology where features present in images can be extracted from the matching reports. Labeling medical reports can be very time-consuming and, depending on the context, substantial efforts may be required even to create relatively small datasets. Furthermore, many pathologies have a low prevalence (e.g., less than 10%) and will result in datasets with highly imbalanced classes. On a large scale, manually performing this type of labeling task is intractable, and automatized methods are therefore required.

Prior work suggests that the automatized labeling of radiology reports in the English language is feasible (Wood et al., 2020), but several questions remain open. Namely, practical applications often suffer from compounded issues, including non-English texts, a small

Table 1: Evaluation metrics of the classifiers. FCD: focal cortical dysplasia, MTS: mesial temporal sclerosis, HA: hippocampal abnormalities

| | F1-score (macro) | | | Balanced Accuracy | | |
|---|---|---|---|---|---|---|
| **Model** | **FCD** | **MTS** | **HA** | **FCD** | **MTS** | **HA** |
| roberta-base-danish (original) | 0.65 | 0.70 | 0.72 | 0.67 | 0.73 | 0.74 |
| roberta-base-danish (pre-trained) | 0.66 | 0.82 | 0.73 | 0.73 | 0.84 | 0.74 |
| xlm-roberta-base (original) | 0.68 | 0.76 | 0.69 | 0.70 | 0.75 | 0.70 |
| xlm-roberta-base (pre-trained) | **0.70** | **0.88** | 0.69 | **0.74** | **0.89** | 0.69 |
| distiluse-base-multilingual-cased-v2 (SetFit) | 0.56 | 0.79 | **0.77** | 0.54 | 0.80 | **0.78** |

number of labeled samples, and class imbalance. These factors can all adversely impact the performance of NLP models in unique ways and a reliable approach to tackle these issues is yet to be determined.

In this work, we focus on a realistic use case, that of labeling radiology reports of magnetic resonance images (MRI) in the Danish language in a cohort of epilepsy patients. Our primary goal is to evaluate the current state-of-the-art of NLP models in this context and provide a comparative baseline for researchers with similar tasks.

## 2. Methods

### 2.1. Dataset

A dataset of 16,899 Danish radiology reports describing the MRI scans of 4,769 patients with ICD-10 code G40* (epilepsy) was obtained. Additionally, a corpus of 1,2 million Danish radiology reports were retrieved in bulk, irrespective of modality and disease, and used for pre-training. Three types of abnormalities relevant to epilepsy were labeled in the MRI reports: focal cortical dysplasia (FCD) (n=1,122), mesial temporal sclerosis (MTS) (n=904), and hippocampal abnormalities (HA) (n=555). Reports with mention of the abnormalities were identified using regular expressions and manually by a medical student (HK) under the supervision of an expert neurologist (LHP). The FCD dataset was also labeled by a second clinician (MP). The FCD and MTS datasets represent cases where the radiologist described a specific feature directly such as the presence or absence of a given pathology, often associated with a degree of certainty. To account for the variable degree of confidence, the prefixes negative, probable, highly probable (only for FCD), and positive were manually appended to the FCD (n=877/86/93/66) and MTS (n=668/104/132) labels. The HA dataset presents a more complex pathology where abnormalities are indirectly described. In this case, reports were labeled as either normal or abnormal (n=157/398). Labeling of the FCD, MTS, and HA datasets took approximately 35, 25, and 20 hours, respectively. Training and test sets were created using 80%/20% splits, with 20% of the training data used for validation.

### 2.2. Natural Language Processing Models

Three approaches were evaluated: transformer models supporting Danish text (roberta-base-danish) and cross-lingual text including Danish (xlm-roberta-base), without and with pre-training on radiology reports, and few-shot learning with sentence transformers (SetFit) using a multilingual model (distiluse-base-multilingual-cased-v2) (Conneau et al., 2020; Liu et al., 2019; Tunstall et al., 2022). Model pre-training was achieved using whole-word masking. Fine-tuning for text classification was performed using a sequence classification head with weighted (binary) cross-entropy loss.

## 3. Results

The agreement (Cohen's kappa) between the two raters for the FCD dataset was 0.83. Table 1 presents the evaluation metrics for the classifiers on the different datasets. Examples of confusion matrices for the FCD dataset are presented in Figure 1.

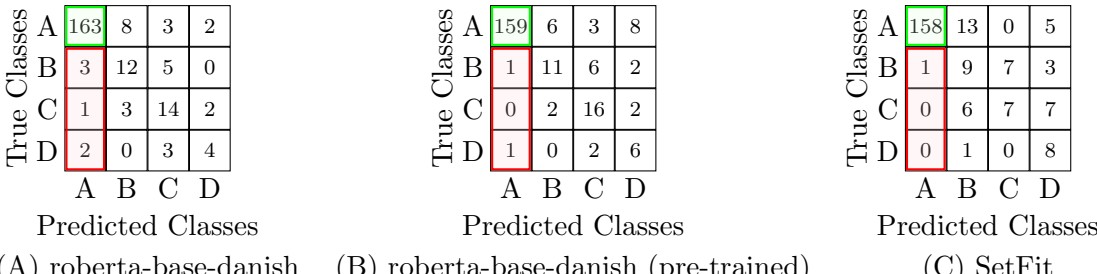

Figure 1: Confusion matrices of selected classifiers on the FCD test dataset. Recall is ■/(■+■). A: No FCD, B: Potential FCD, C: Highly Probable FCD, D: FCD

## 4. Discussion & Conclusion

In this work, we evaluated a range of approaches and datasets representing the task of medical text classification in small and imbalanced datasets in a non-English Language.

Original transformers models were evaluated to account for the scenario where a large corpus in the target domain and language may not be readily available for fine-tuning. In general, the availability of transformers pre-trained for specific domains in non-English languages is a core issue for the generalizability of NLP approaches. In English, prior work indicates that larger categories of brain abnormalities could be reliably labeled from radiology reports (0.93-0.99% accuracy). However, a similar approach applied to a Polish dataset yielded F1-scores between 0.75 and 0.81 (Obuchowski et al., 2023). Although these studies are admittedly different, in combination with our results they do suggest a worrying trend of reduced performance in less-represented languages.

By including a model natively trained on a Danish corpus (roberta-base-danish) and a cross-lingual model (xlm-roberta-base), we sought to compare a model trained on a small unilingual corpus and a cross-lingual model trained on a larger corpus. Our results indicate that the latter offers better performance for our specific task. However, we note that roberta-base-danish may be underpowered compared to more recent models and future work should consider additional encoders. Our evaluation provides a point of reference showcasing the possible gain in performance provided by pre-training. As expected, it did improve performance in almost all cases, however, it is important to emphasize that obtaining a relevant corpus may be non-trivial and can require substantial time and/or resources.

SetFit has been recently introduced as a competitive approach to the traditional transformers for small datasets. Interestingly, in our scenario, this approach both achieved the worst and best performances on the FCD and HA datasets, respectively.

It is important to emphasize that none of the models evaluated here exhibit performances sufficient to provide a reliable and fully automated solution. However, a closer look at the confusion matrices reveals that some of the classifiers have an almost perfect recall for the most numerous class (Fig. 1B-C). Therefore, when manually labeling large datasets a substantial amount of work could potentially be avoided by first using the classifier to identify the reports belonging to that class and then only processing the remainder. However, the performance of this approach is heavily dependent on the dataset and would have to be carefully validated.

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
