# OpenReview forum: "Classification of Medical Text in Small and Imbalanced Datasets in a Non-English Language"
_MIDL.io/2024/Short_Papers — MIDL 2024 Short Papers_

### Official Review · Reviewer_QBND · 2024-04-24

**Confidence:** 3
**Final Rating:** 3.5

**Review:**

The authors use a large non-disease specific corpus of radiology reports in Danish to pretrain (or not) language models. They label manually several thousand reports for epilepsy-related abnormalities read from MRI images. They then benchmark several language models for classificaiotn.

Domain-specific LLMs, such as BioBERT for biomedical data, can perform much better than general LLMs. This should be attempted. If not biomedical-specific LLM exists in Danish, it would be of interest to try a simple approach by translating reports into English and then applying a domain-specific LLM to the task.

Strengths
- large pre-training corpus and disease-specific cohort
- several  models benchmarked

Weaknesses
- inter-rater agreement should be provided, at least on a subsect of cases. this can be quite low and can help put performance of LLMs in context.
- lacking discussion of low improvement between original and pretrain in FCD and HA classs, but high improvement in MTS class. data size is about the same between FCD and MTS, so this is surprising, maybe due to different labelling process?

---

### Decision · Program_Chairs · 2024-04-26

Accept